# Turntable IMU Calibration Algorithm Based on the Fourier Transform Technique

**DOI:** 10.3390/s23021045

**Published:** 2023-01-16

**Authors:** Yury Bolotin, Vladimir Savin

**Affiliations:** Mathemaitcs and Mechanics Department, Moscow Lomonosov State University, Moscow 119991, Russia

**Keywords:** IMU, calibration, Fourier transform, FFT

## Abstract

The paper suggests a new approach to calibration of a micromechanical inertial measurement unit. The data are collected on a simple rotating turntable with horizontal (or close to) rotation axis. For such a turntable, an electric screwdriver with fairly low rotation rate can be used. The algorithm is based on the Fourier transform applied to the rotation experimental data, implemented as FFT. The frequencies and amplitudes of the spectral peaks are calculated and collected in a small set of data, and calibration is done explicitly with these data. Calibration of an accelerometer triad and choosing the IMU coordinate frame are reduced to approximating the collected data with an ellipsoid in three dimensions. With rotation frequency calculated as the peak frequency of accelerometer readings, calibration of the gyros is a straightforward linear least square problem. The algorithm is purely algebraic, requires no iterations and no initial guess on the parameters, and thus encounters no convergence problems. The algorithm was tested both with simulated and experimental data, with some promising results.

## 1. Introduction

Calibration of an inertial measurement unit (IMU) is a necessary step in preparing the IMU for operation, that being either inertial navigation or some supplementary task in biomechanics or robotics. The purpose of calibration is to provide formulas and estimate parameters needed to convert the raw accelerometer and gyro readings into the specific force and the angular rate of the unit. The majority of calibration models found in the literature are linear and include scale factors and biases; similar models are used in this paper. The calibration procedure consists of experiments with the IMU and algorithms to process the collected data.

Calibration of an IMU is fairly straightforward if done on a precise turntable with two or three degrees of freedom, measuring rotation angles with high accuracy. The table is usually well-calibrated, and its azimuth angle relative to the North is known, thus the true angular velocity and specific force are also known. In such a case, the calibration procedure usually consists of several steps—different static positions and rotations with prescribed angular velocities. Therefore, the data processing algorithms consist of several respective steps to calibrate biases and scale factors of accelerometers and angular velocity sensors. Mathematical formulas for these steps are often straightforward and can be reduced to linear least squares [1,2], or Kalman filtering [3,4].

A series of papers considers IMU calibration when a so-called low-grade turntable [5] is used. Such a turntable delivers rotation, but either it does not provide information about rotation rate or angles, or this information is inaccurate. Thus, the only information available is the data taken from the IMU sensors. In such settings, IMU calibration cannot be done step-by-step, and dynamic equations of motion must be taken into account. In Refs. [3,5], an algorithm for calibration on a low-grade turntable based on the extended Kalman filter was suggested, with very good results, including applications for calibrating both low-cost and tactical grade IMUs. The Kalman filter algorithm is optimal under the well-known more or less reasonable assumptions, and is easy to implement. Unfortunately, in practice, to converge, it requires a fairly good initial guess on the IMU orientation and scale factors. Another difficulty arises from the necessity to know the IMU sensors noises to obtain the optimal results.

When we consider calibrating low-cost micromechanical IMUs, which are now broadly used, in, for instance, biomechanics, robotics or pedestrian dead reckoning (see earlier work [6], later [7,8], or recent [9,10,11]), the calibration procedure should be straightforward and reliable. In most practical cases, using a precise turntable is impossible. A lot of approaches to calibration of micromechanical IMUs without special equipment such as a turntable are known, see, for example, earlier work [12], or more recent [13,14,15]. For calibrating accelerometers, the IMU is put to several (about eighteen) static positions. For calibrating gyros, the IMU is rotated from one such position to another by hand. With collected data, the calibration problem is set as a maximum likelihood optimization problem, which is usually solved by numerical optimization. Note that there is one drawback of IMU calibration without putting it on a turntable: it is difficult to produce consistent regular rotations by hand; thus, the angular velocity sensors scale factors and misalignments can be estimated inaccurately. Another drawback lies with numerical optimization: it requires an initial guess and can diverge if such a guess is not good.

Having in mind the argument above, we think that using some turntable for calibration, preferably the simplest one, is advisable. In a recent paper [16], a thorough procedure to calibrate a low-cost IMU on a rotation bench is proposed, with excellent experimental verification. The procedure allows to estimate not just linear, but also a nonlinear calibration model. It should be noted, however, that the algorithms are based on numerical optimization of the cost function and thus require an initial guess on some parameters.

This paper proposes an approach to calibration of a micromechanical IMU using either a low-grade turntable, or just a common home tool such as an electric screwdriver. Such a screwdriver often allows two speeds of rotation and reverse rotation. The proposed calibration experiment is as follows. The screwdriver is rigidly mounted on a workbench with the rotation axis approximately horizontal. The IMU is fixed in some position to the screwdriver shaft and is rotated for several tens of revolutions. Then, the position of the IMU on the shaft is changed, and revolutions are repeated, and so on. The number of positions must not be less than three, the positions must differ from one another by turns around axes approximately perpendicular to the shaft. The screwdriver orientation, angular velocity or the number of revolutions are not measured otherwise but by the IMU sensors. The screwdriver rotation can be nonuniform due to imperfections of the motor or gearbox, but must be monotonic with approximately constant duration of one revolution.

The proposed data processing algorithm first applies FFT (fast Fourier transform) to obtain the Fourier transform of the sensor data. The next step is to localize the frequencies corresponding to the peaks of the spectrum, and to calculate the amplitudes and phases of the spectrum at the peaks. The peaks are at zero frequency, at screwdriver revolution frequency, and maybe at some higher frequencies due to imperfections of the screwdriver gearbox. The frequencies, amplitude and phase data are collected in a small data array. The subsequent steps of the algorithm work with this data array, and are purely algebraic.

First, the accelerometers are calibrated. We apply here the well-known idea of fitting the raw accelerometer data with an ellipsoid [17]. The only difference from [17] is that, here, the ellipsoid lies in the space of spectral coefficients, not in the physical space of accelerometer readings. Since the ellipsoid-fitting algorithm is purely algebraic, it does not require any initial guess, and the convergence issues do not arise. Along with calibrating the accelerometers, the algorithm selects the instrument coordinate frame and computes the angular velocities of rotation in the instrument frame. Provided that the instrument frame is selected, the mean angular velocity of revolutions is computed and the accelerometers are calibrated, the angular rate sensor calibration is reduced to a straightforward linear algebraic problem.

We have not found many papers about IMU calibration by applying the Fourier technique to the turntable experiment. One such paper is Ref. [18]. The calibration model in Ref. [18] is the same as ours. However, the big difference is that in Ref. [18], the true angular velocity and the true specific force are assumed known from the turntable data. With this assumption, the equations are completely linear and can be solved without iterations both in the time domain and in the spectral domain. In our case where the true sensors readings are unknown, the initial equations are nonlinear, thus, as far as we know, non-iterative algorithms in the time domain do not exist, which leaves the spectral approach we propose the only non-iterative one. Some preliminary results were published by us in Ref. [19].

The paper is organized as follows. First, we state the problem. Next, the necessary mathematical formulas are derived and analyzed. Further on, we present results of simulation and experiments with a micromechanical IMU that support our reasoning. The bold letters such as f denote column vectors in the three-dimensional space, the bold capital letters such as F denote column vectors in the three-dimensional space of Fourier transforms. When the coordinate frame the column vectors are written in is of importance, we write them as $f^{.z}$, where $z_{1}z_{2}z_{3}$ are the coordinate frame axes.

## 2. Calibration Problem Statement

### 2.1. Calibration Model

The IMU measures the angular rate ω of the sensor relative to the inertial frame and the specific force f acting on the proof mass *M* of the accelerometer. In some cases, especially for IMU operating in the environment with high angular rates, it is necessary to assume several proof masses. However, here, we assume just one proof mass. Both the angular rate and the specific force are given by the column vectors of their coordinates $\omega^{z}$ and $f^{z}$ in the so-called instrument frame $Mz_{1}z_{2}z_{3}$. The origin of this frame is at the proof mass, its axes $z_{1},z_{2},z_{3}$ are rigidly connected with the instrument body.

Under the linear calibration model, the raw accelerometer data a(t) and the raw gyro data w(t) are connected with the true specific force $f^{.z}$ and angular velocity $\omega^{.z}$ written in the instrument frame $Mz_{1}z_{2}z_{3}$ by the calibration formulas [3,7]: (1)$$\begin{array}{ccc} f^{.z}\left(t\right) & = & S_{f}a\left(t\right)+b_{f}+\delta f\left(t\right), \end{array}$$(2)$$\begin{array}{ccc} \omega^{.z}\left(t\right) & = & S_{\omega }w\left(t\right)+b_{\omega }+\delta \omega \left(t\right), \end{array}$$
where δf(t),δω(t) are the unmodeled measurement errors. The goal of calibration is to estimate the 3×3 matrices $S_{f},S_{\omega }$ and the 3×1 column vectors $b_{f},b_{\omega }$. Many calibration procedures can be found in the literature; some are mentioned in the introduction section. In this paper, we consider the following calibration procedure settings.

The IMU is mounted on the shaft of a turntable or a screwdriver in several positions and, in each position, it is rotated for several tens of revolutions using the screwdriver motor. These rotations are called “experiments”. The screwdriver body does not move relative to the Earth during all the experiments.

We use the following coordinate frames (Figure 1). The navigation frame $On_{1}n_{2}n_{3}$ is connected with the Earth, its axis $n_{3}$ points up, the axis $n_{1}$ is the projection of the shaft rotation axis on the horizon. The axes origin *O* lies on the shaft rotation axis. The azimuth angle between $n_{1}$ and the East direction is denoted as ψ. The frame $Os_{1}s_{2}s_{3}$ is rigidly tied to the body of the screwdriver, its $s_{1}$ axis is directed along the shaft, the angle 0<α<π/2 of this axis with the horizon is constant. The axis $s_{3}$ lies in the vertical plane, the axis $s_{2}$ lies in the horizontal plane. The frame $Oe_{1}e_{2}e_{3}$ is rigidly tied to the shaft: the axis $Oe_{1}$ points along the shaft, the axes $Oe_{2},Oe_{3}$ rotate in the plane perpendicular to the shaft during experiments. The IMU instrument frame $Mz_{1}z_{2}z_{3}$ is rigidly tied to the shaft in each experiment but changes its orientation relative to the shaft from experiment to experiment. The proof mass *M* of the accelerometer can lie out of the shaft rotation axis: the centrifugal forces induced by the shaft rotation are accounted for by the algorithm.

### 2.2. Experiments and Data Models

Let *P* be the number of experiments (different rotations) numbered by p=1,…,P: in each experiment, the shaft axes $e_{2},e_{3}$ rotate around the axis $e_{1}$, the angle of the axis $e_{2}$ relative to the stand axis $s_{2}$ is denoted by $\theta_{p}\left(t\right)$:(3)$$\theta_{p}\left(t\right)=\omega_{p}t+\gamma_{p}+\delta \theta_{p}\left(t\right),\;-T_{p}/2\leq t\leq T_{p}/2.$$
Here, $T_{p}$ is the duration of the experiment, $\omega_{p}$ is the angular velocity of the shaft, $\gamma_{p}$ is the phase shift, $\delta \theta_{p}\left(t\right)$ is the disturbance due to imperfection of the turntable. For brevity, we set $\delta \theta_{p}\left(t\right)$ to zero in the theoretical section of this paper (influence of this term can be compensated for by analyzing the higher frequency spectrum of the sensors). Let us complement experiments—rotations with static experiments were the IMU stands still at different positions, denoting these experiments with numbers q=1,…,Q:(4)$$\theta_{q}\left(t\right)=\gamma_{q},\;-T_{q}/2\leq t\leq T_{q}/2.$$

Let g and u be the gravity and the Earth angular velocity vectors, *g* and *u* be their absolute values. In the navigation frame, these vectors are written as
(5)$$g=-gn_{3},\;u=n_{1}\sin\psi \cos\phi +n_{2}\cos\psi \cos\phi +n_{3}\sin\phi ,$$
where ϕ is the latitude, ψ is the azimuth angle of the screwdriver shaft. In the screwdriver frame $Os_{1}s_{2}s_{3}$, these vectors can be written as
(6)$$\begin{array}{cc} -g & =g_{1}s_{1}+g_{2}s_{3},\\ u & =u_{1}s_{1}+u_{2}s_{2}+u_{3}s_{3}. \end{array}$$
Here, we use the notation
(7)$$\begin{array}{cc} g_{1} & =g\sin\alpha ,\;g_{2}=g\cos\alpha ,\\ u_{1} & =\cos\alpha \sin\psi \cos\phi +\sin\alpha \sin\phi ,\;u_{2}=\cos\psi \cos\phi ,\\ u_{3} & =\cos\alpha \sin\phi -\sin\alpha \sin\psi \cos\theta . \end{array}$$

Note that ψ,α are unknown before calibration, thus the constants in (7) are also unknown. In the shaft frame, the gravity g and the Earth angular velocity u during the *p*-th experiment can be written as
(8)$$\begin{array}{cc} -g_{p}\left(t\right) & =g_{1}e_{1}+g_{2}\left[e_{2p}\sin\theta_{p}\left(t\right)+e_{3p}\cos\theta_{p}\left(t\right)\right],\\ u_{p}\left(t\right) & =u_{1}e_{1}+e_{2p}\left[u_{2}\cos\theta_{p}\left(t\right)+u_{3}\sin\theta_{p}\left(t\right)\right]+e_{3p}\left[-u_{2}\sin\theta_{p}\left(t\right)+u_{3}\cos\theta_{p}\left(t\right)\right]. \end{array}$$

Now, we remember that the sensor angular velocity and the specific force acting on the proof mass in the *p*-th experiment can be written as
(9)$$\begin{array}{cc} \omega_{p}\left(t\right) & =u_{p}\left(t\right)+{\dot{\theta }}_{p}\left(t\right)e_{1},\\ f_{p}\left(t\right) & =-g_{p}\left(t\right)+\omega_{p}\left(t\right)\times \omega_{p}\left(t\right)\times \rho_{p}, \end{array}$$
where $\rho_{p}=\mathrm{const}\;$ is the vector of displacement of the proof mass relative to the shaft. In the instrument frame in the *p*-th experiment, the above vectors can be written as
(10)$$f_{p}\left(t\right)=g_{1}e_{1p}+g_{2}\left[e_{2}\sin\theta_{p}\left(t\right)+e_{3}\cos\theta_{p}\left(t\right)\right]-{\dot{\theta }}_{p}^{2}\left[e_{2p}\rho_{2p}+e_{3p}\rho_{3p}\right],$$
(11)$$\begin{array}{cc} \omega_{p}\left(t\right) & =\left[u_{1}+{\dot{\theta }}_{p}\left(t\right)\right]e_{1}+\\ \; & +\left[u_{2}\cos\theta_{p}\left(t\right)+u_{3}\sin\theta_{p}\left(t\right)\right]e_{2p}+\left[-u_{2}\sin\theta_{p}\left(t\right)+u_{3}\cos\theta_{p}\left(t\right)\right]e_{3p}. \end{array}$$
Here, $\rho_{1p},\rho_{2p},\rho_{3p}$ are the components of $\rho_{p}$ along the $e_{1p},e_{2p},e_{3p}$ axes. Writing down the centrifugal force, we have neglected its component due to the Earth rotation.

Turning $e_{3p},e_{2p}$, $e_{3q},e_{2q}$ appropriately in the plane perpendicular to the shaft, we can assume without loss of generality that $\gamma_{p}=0$, $\gamma_{q}=0$ in (3), (4). Thus, we can write $\theta_{p}\left(t\right)=\omega_{p}t+\delta \theta_{p}\left(t\right)$, $\theta_{q}=0$.

For the static experiments (4), where the screwdriver motor is switched off, the IMU measurements can be written as
(12)$$\begin{array}{cc} f_{q} & =g_{1}e_{1q}+g_{2}e_{3q}, \end{array}$$
(13)$$\begin{array}{cc} \omega_{q} & =u_{1}e_{1q}+u_{2}e_{2q}+u_{3}e_{3q}. \end{array}$$

Equations (10)–(13) were written in the invariant vector form. Further on, we assume that they are written as column vectors in the instrument frame: $f=f^{.z}$, etc. However, since the instrument frame is the only frame used below, we omit the superscript .z. Note that the column vectors $e_{1p},e_{2p},e_{3p}$ and $e_{1p},e_{2p},e_{3p}$ written in the instrument frame are constant during each experiment but change from experiment to experiment.

The calibration task is to estimate $S_{f},b_{f},S_{\omega },b_{\omega }$ from (10), (12), and from (11), (13) for calibration models (1), (2). The required mathematical formulas are discussed in the next section.

## 3. Calibration Algorithm

This section covers mathematical foundations of the proposed algorithm; some technical details of implementation are discussed in the results section.

### 3.1. Fourier Transform for Accelerometer Calibration Formulas

In this section, we neglect imperfections of the motor, setting $\delta \theta_{p}\left(t\right)=0$, thus setting $\theta_{p}\left(t\right)=\omega_{p}t$. Note that these imperfections do not influence the gyro calibration due to their high-frequency spectrum, but can cause a bias in accelerometer calibration due to the centrifugal force bias. However, these biases can be calculated by analyzing the oscillations in the angular velocity measurements. We do not pursue this matter further here.

Let us apply the Fourier transform [20] in (1), (10) and denote the Fourier images as $f_{p}\left(t\right)\;\to \;F_{p}\left(\omega \right)$, etc.:
(14)$$f_{p}\left(t\right)\;\to \;F\left\{f_{p}\left(t\right)\right\}=F_{p}\left(\omega \right)=\frac{1}{2\pi }\int_{-\infty }^{\infty }e^{-i\omega t}f_{p}\left(t\right)\;dt.$$

The Fourier transform in (1) for both rotations and static experiments takes the form
(15)$$\begin{array}{c} F_{p}\left(\omega \right)=S_{f}A_{p}\left(\omega \right)+\delta \left(\omega \right)b_{f}+\delta F_{p}\left(\omega \right),\;p=1,\ldots ,P, \end{array}$$
(16)$$\begin{array}{c} F_{q}\left(\omega \right)=S_{f}A_{q}\left(\omega \right)+\delta \left(\omega \right)b_{f}+\delta F_{q}\left(\omega \right),\;q=1,\ldots ,Q, \end{array}$$
where δ(ω) is the delta-function. The Fourier transform in (10) takes the form
(17)$$\begin{array}{cc} F_{p}\left(\omega \right) & =\left[g_{1}e_{1p}-\omega_{p}^{2}\rho_{2p}e_{2p}-\omega_{p}^{2}\rho_{3p}e_{3p}\right]\delta \left(\omega \right)+\\ \; & +\frac{g_{2}}{2}\left[ie_{2p}+e_{3p}\right]\delta \left(\omega -\omega_{p}\right),+\frac{g_{2}}{2}\left[-ie_{2p}+e_{3p}\right]\delta \left(\omega +\omega_{p}\right) \end{array}$$
(18)$$\begin{array}{cc} F_{q}\left(\omega \right) & =\left[g_{1}e_{1q}+g_{2}e_{3q}\right]\delta \left(\omega \right). \end{array}$$

We have used the following well-known Fourier transform formulas [20]:(19)$$\begin{array}{cccccc} \; & c=\mathrm{const}\; & \to & \; & & c\delta (\omega ),\\ \; & \sin\theta_{p}\left(t\right)=\sin\left(\omega_{p}t\right) & \to & \; & - & \frac{i}{2}\delta \left(\omega -\omega_{p}\right)+\frac{i}{2}\delta \left(\omega +\omega_{p}\right),\;\;i=\sqrt{-1},\\ \; & \cos\left(\omega_{p}t\right)=\cos\left(\omega_{p}t\right) & \to & \; & \; & \frac{1}{2}\delta \left(\omega -\omega_{p}\right)+\frac{1}{2}\delta \left(\omega +\omega_{p}\right). \end{array}$$

Comparing (17), (18) and (15), (16), we see that the Fourier transform $A_{p}\left(\omega \right)$ has delta-function-type peaks at $\omega =0,\omega =\pm \omega_{p}$, while the Fourier transform $A_{q}\left(\omega \right)$ has delta-function-type peaks at ω=0. Let us denote the amplitudes of these peaks as
(20)$$\int_{-\epsilon }^{\epsilon }A_{p}\left(\omega \right)\;d\omega ={\bar{A}}_{p},\;\int_{\pm \omega_{p}-\epsilon }^{\pm \omega_{p}+\epsilon }A_{p}\left(\omega \right)\;d\omega =\frac{1}{2}\left(R_{p}\pm I_{p}\right),\;\int_{-\epsilon }^{\epsilon }A_{q}\left(\omega \right)\;d\omega ={\bar{A}}_{q}.$$

The Fourier transforms $A_{p}\left(\omega \right)$, $A_{p}\left(\omega \right)$ take the form
(21)$$\begin{array}{cc} A_{p}\left(\omega \right) & ={\bar{A}}_{p}\delta \left(\omega \right)+R_{p}\left[\delta \left(\omega -\omega_{p}\right)+\delta \left(\omega +\omega_{p}\right)\right]+iI_{p}\left[\delta \left(\omega -\omega_{p}\right)-\delta \left(\omega +\omega_{p}\right)\right],\\ A_{q}\left(\omega \right) & ={\bar{A}}_{q}\delta \left(\omega \right). \end{array}$$

Note that $\omega_{p}$ here can be calculated as the position of a peak of $A_{p}\left(\omega \right)$ in the interval ω>0, if the rotation was clockwise, or in the interval ω<0, if the rotation was counter-clockwise. Equations (15) and (16) can now be rewritten as
(22)$$\begin{array}{cc} F_{p}\left(\omega \right) & =\left(S_{f}{\bar{A}}_{p}+b_{f}\right)\delta \left(\omega \right)+S_{f}\left(R_{p}+iI_{p}\right)\delta \left(\omega -\omega_{p}\right)+S_{f}\left(R_{i}-iI_{p}\right)\delta \left(\omega +\omega_{p}\right)+\delta F\left(\omega \right),\\ F_{q}\left(\omega \right) & =\left(S_{f}{\bar{A}}_{q}+b_{f}\right)\delta \left(\omega \right). \end{array}$$

Collecting coefficients at delta-functions in the equation above and in (17), (18), we obtain
(23)$$\begin{array}{ccc} g_{1}e_{1q}+g_{2}e_{3q} & = & S_{f}{\bar{A}}_{q}+b_{f},\\ g_{1}e_{1p}+g_{2}e_{3p}-\omega_{p}^{2}\rho_{2p}e_{2p}-\omega_{p}^{2}\rho_{3p}e_{3p} & = & S_{f}{\bar{A}}_{p}+b_{f},\\ g_{2}e_{3p} & = & S_{f}R_{p},\\ g_{2}e_{2p} & = & S_{f}I_{p}. \end{array}$$

Equation (23) are the base for calibrating the accelerometers. They can be resolved in the unknowns $S_{f},b_{f},g_{1},g_{2},e_{1p},e_{2p},e_{1p},e_{2p}$ in several ways. We take the simplest way here, not claiming it to be the most accurate one.

Let us exclude the unknown constants $g_{1},g_{2},e_{1p},e_{2p},e_{3p},e_{1q},e_{2q}$ from (23) by algebraic manipulations:
(24)$$\begin{array}{ccccc} g^{2} & = & g_{1}^{2}e_{1q}^{T}e_{1q}+g_{2}^{2}e_{2q}^{T}e_{2q} & = & {\bar{A}}_{q}^{T}S_{f}^{T}S_{f}{\bar{A}}_{q}+2{\bar{A}}_{q}^{T}S_{f}^{T}b_{f}+b_{f}^{T}b_{f},\\ 0 & = & g_{2}^{2}e_{1p}^{T}e_{2p} & = & {\hat{I}}_{p}^{T}S_{f}^{T}S_{f}{\hat{R}}_{p},\\ 0 & = & g_{2}^{2}\left(e_{2p}^{T}e_{2p}-e_{3p}^{T}e_{3p}\right) & = & {\hat{I}}_{p}^{T}S_{f}^{T}S_{f}{\hat{I}}_{p}-{\hat{R}}_{p}^{T}S_{f}^{T}S_{f}{\hat{R}}_{p}. \end{array}$$

Equation (24) can be resolved to find $S_{f},b_{f}$. Note that the choice of the sensor frame is arbitrary provided it is fixed in the instrument body; thus, $S_{f}$ includes only six independent variables [3]. For a given number P,Q of experiments, we have Q+2P equations. It looks like that to calculate 6 + 3 = 9 calibration parameters, it suffices to set Q=3,P=3. However, a more careful look shows that we must take Q≥4, see below.

### 3.2. Fitting Accelerometer Data to an Ellipsoid

Equation (24) is quadratic. To reduce these to linear equations, we use the well-known trick [17]: introduce the symmetric positive matrix *M*, the vector ***m***, and the constant *m*_0_ as
(25)$$\begin{array}{c} M=S_{f}^{T}S_{f},\;m=S_{f}^{T}b,\;m_{0}^{2}=b^{T}b=m^{T}M^{-1}m. \end{array}$$

Equation (24) can then be rewritten as
(26)$$\begin{array}{ccc} g^{2} & = & {\bar{A}}_{q}^{T}M{\bar{A}}_{q}+2m^{T}{\bar{A}}_{q}+m_{0}^{2},\\ 0 & = & I_{p}^{T}MR_{p},\\ 0 & = & I_{p}^{T}MI_{p}-R_{p}^{T}MR_{p}. \end{array}$$

Equation (26) defines an ellipsoid in the three-dimensional space. To obtain its parameters, let us divide the first equation in (26) with $g^{2}-m_{0}^{2}$, and use the notation
(27)$$\bar{M}=\frac{1}{g^{2}-m_{0}^{2}}M,\;\bar{m}=\frac{1}{g^{2}-m_{0}^{2}}m.$$

We get
(28)$$\begin{array}{cc} 1= & {\bar{A}}_{q}^{T}\bar{M}{\bar{A}}_{q}+2{\bar{m}}^{T}{\bar{A}}_{q},\\ 0= & I_{p}^{T}\bar{M}R_{p},\\ 0= & I_{p}^{T}\bar{M}I_{p}-R_{p}^{T}\bar{M}R_{p}. \end{array}$$

Note that if Q=3, and the column vectors ${\bar{A}}_{1},{\bar{A}}_{2},{\bar{A}}_{3}$ are linearly independent, then the system (28) allows the false trivial solution
(29)$$\bar{M}=0,\;\bar{m^{T}}=\frac{1}{2}\left[\begin{array}{ccc} 1 & 1 & 1 \end{array}\right]\cdot {\left[\begin{array}{ccc} {\bar{A}}_{1} & {\bar{A}}_{2} & {\bar{A}}_{3} \end{array}\right]}^{-1},$$
which will follow with the estimate of the scaling matrix as $S_{f}=0$. Thus, for reliable calibration, we must set Q≥4.

Expanding coefficients for $\bar{M},\bar{m}$ to a vector as
(30)$$\begin{array}{c} \mathrm{Row}\;\left(A,B\right)=\left[\begin{array}{cccccc} A_{1}B_{1} & A_{2}B_{2} & A_{3}B_{3} & A_{1}B_{2}+A_{2}B_{1} & A_{1}B_{3}+A_{3}B_{1} & A_{2}B_{3}+A_{3}B_{2} \end{array}\right], \end{array}$$
and expanding the unknowns to the vector
(31)$$\;\begin{array}{c} x={\left[\begin{array}{ccccccccc} {\bar{M}}_{11} & {\bar{M}}_{22} & {\bar{M}}_{33} & {\bar{M}}_{12} & {\bar{M}}_{13} & {\bar{M}}_{23} & {\bar{m}}_{1} & {\bar{m}}_{2} & {\bar{m}}_{3} \end{array}\right]}^{T}, \end{array}$$
we obtain the system of linear equations in *x*
(32)$$\begin{array}{c} \begin{array}{cccc} 1 & = & [\mathrm{Row}\;\left({\bar{A}}_{q},{\bar{A}}_{q}\right)\;2{\bar{A}}_{q}^{T}]\cdot x,\;\;\;\; & q=1,\ldots ,Q\\ 0 & = & [\mathrm{Row}\;\left(I_{p}^{T},R_{p}\right)\;0_{1\times 3}]\cdot x,\;\;\;\; & p=1,\ldots ,P,\\ 0 & = & [\mathrm{Row}\;\left(I_{p}^{T},I_{p}\right)-\mathrm{Row}\;\left(R_{p}^{T},R_{p}\right)\;0_{1\times 3}]\cdot x.\; & \end{array} \end{array}$$

The overdetermined system (32) can be solved in the mean square sense to obtain *x*, then $\bar{M}$,
$\bar{m}$ from (31), and further on
(33)$$M=\frac{g^{2}}{1+{\bar{m}}^{T}{\bar{M}}^{-1}\bar{m}}\bar{M},\;m=\frac{g^{2}}{1+{\bar{m}}^{T}{\bar{M}}^{-1}\bar{m}}\bar{m}.$$

Finally, the calibration parameters for the accelerometers $S_{f}$, $b_{f}$, are obtained by factorizing *M* in upper triangular, lower triangular or symmetric form, depending on the situation:
(34)$$M=S_{f}^{T}S_{f},\;b_{f}=S_{f}^{-T}m.$$

Knowing the calibration parameters, we can find *g*_1_, *g*_2_, α from (23) as:
(35)$$g_{2}=\sqrt{\frac{1}{2P}\sum_{p=1}^{P}\left[∥S_{f}R_{p}{∥}^{2}+{∥S_{f}I_{p}∥}^{2}\right]},\;g_{1}=\sqrt{g^{2}-g_{2}^{2}},\;\alpha =\mathrm{asin}\frac{g_{1}}{g}.$$

Next, we calculate the orts $e_{1p}$, $e_{2p}$, $e_{3p}$ as column vectors in the instrument frame as:
(36)$$e_{3p}=\frac{1}{g_{2}}S_{f}R_{p},\;e_{2p}=\frac{1}{g_{2}}S_{f}I_{p},\;e_{1p}=e_{2p}\times e_{3p}.$$

Now, we have to discuss one question. If the directions of rotation are known beforehand, then the signs of $\omega_{p}$ are also known, and the formulas (36) are quite correct. Now suppose that we do not know these signs, and have taken the wrong sign of some $\omega_{p}$. Then, the sign of $I_{p}$ will be wrong; consequently, the signs of $e_{1p}$, $e_{2p}$ will be also wrong. To find the correct sign, let us multiply the second equation in (23) with $e_{1p}^{T}$ taken from (36): we get $g_{1}=e_{1}^{T}\left(S_{f}{\bar{A}}_{p}+b_{f}\right)$. We know that the value *g*_1_ = *g* sin α is positive; thus, the scalar product on the right must be also positive. If it is negative, we must change the sign of $\omega_{p}$ and change the signs of $e_{1p}$, $e_{2p}$ before proceeding to calibrate the gyros.

Note that the vectors $e_{1q}$, $e_{2q}$, $e_{3q}$ cannot be explicitly calculated from the above equations. However, if the experiments were done in such a way that each *q*-th static experiment precedes to or follows after some *p*-th rotation experiment without dismounting the IMU from the screwdriver shaft, then we can set $e_{1q}$ = $e_{2q}$. However, we cannot do the same with $e_{2q}$, $e_{3q}$, because the rotation angle of these unit vectors around the shaft can be different from the rotation angles of $e_{2q}$, $e_{3q}$.

### 3.3. Calibration of Angular Rate Sensors

When calibrating accelerometers, we have obtained the axes $e_{1p},e_{2p},e_{3p}$ of the shaft frame in the sensor frame. We have also obtained the rotation angular rates $\omega_{p}$ as frequencies of the peaks of the Fourier transform of the accelerometer data.

To calculate the gyros calibration parameters, let us perform the Fourier transform in (11), (13): $\omega_{p}\left(t\right)\to \Omega_{p}\left(\omega \right)$, $w_{p}\left(t\right)\to W_{p}\left(\omega \right)$, to obtain
(37)$$\begin{array}{ccc} \Omega_{p}\left(\omega \right) & = & S_{\omega }W_{p}\left(\omega \right)+2\pi b_{\omega }\delta \left(\omega \right)+\delta \Omega_{p}\left(\omega \right),\\ \Omega_{q}\left(\omega \right) & = & S_{\omega }W_{q}\left(\omega \right)+2\pi b_{\omega }\delta \left(\omega \right)+\delta \Omega_{q}\left(\omega \right). \end{array}$$
Here,
(38)$$\begin{array}{ccc} \Omega_{q}\left(\omega \right)= & \left(u_{1}e_{1q}+u_{2}e_{2q}+u_{3}e_{3q}\right)\delta \left(\omega \right), & \\ \Omega_{p}\left(\omega \right)= & \left(u_{1}+\omega_{p}\right)e_{1p}\delta \left(\omega \right)+ & \\ \;+ & \frac{1}{2}\left[\left(u_{2}e_{2p}+u_{3}e_{3p}\right)+i\left(u_{2}e_{3p}-u_{3}e_{2p}\right)\right]\delta \left(\omega -\omega_{p}\right)+ & \\ \;+ & \frac{1}{2}\left[\left(u_{2}e_{2p}+u_{3}e_{3p}\right)-i\left(u_{2}e_{3p}-u_{3}e_{2p}\right)\right]\delta \left(\omega +\omega_{p}\right). \end{array}$$

As earlier mentioned, introducing the peak values of the angular velocity spectrum
(39)$$\int_{-\epsilon }^{\epsilon }W_{p}\left(\omega \right)\;d\omega ={\bar{W}}_{p},\;\int_{\pm \omega_{p}-\epsilon }^{\pm \omega_{p}+\epsilon }W_{p}\left(\omega \right)\;d\omega =\frac{1}{2}\left(R_{p}\pm I_{p}\right),\;\int_{-\epsilon }^{\epsilon }W_{q}\left(\omega \right)\;d\omega ={\bar{W}}_{q},$$
we can write
(40)$$W_{p}\left(\omega \right)={\bar{W}}_{p}\delta \left(\omega \right)+\left(R_{p}+iI_{p}\right)\delta \left(\omega -\omega_{p}\right)+\left(R_{p}-iI_{p}\right)\delta \left(\omega +\omega_{p}\right).$$

When we collect factors before the delta-functions in (37), (38), we obtain the system of 9P+3Q equations
(41)$$\begin{array}{ccc} u_{1}e_{1q}+u_{2}e_{2q}+u_{3}e_{3q} & = & S_{\omega }{\bar{W}}_{q}+b_{\omega },\;q=1,\ldots ,Q\\ \left(u_{1}+\omega_{p}\right)e_{1p} & = & S_{\omega }{\bar{W}}_{p}+b_{\omega }, \end{array}$$
(42)$$\begin{array}{ccc} u_{2}e_{2p}+u_{3}e_{3p} & = & S_{\omega }R_{p},\;\;\;p=1,\ldots ,P,\\ u_{2}e_{3p}-u_{3}e_{2p} & = & S_{\omega }I_{p}. \end{array}$$

We have 21 unknowns here: besides the twelve calibration parameters $S_{\omega },b_{\omega }$, we do not know nine more: $u_{1},u_{2},u_{3}$, $e_{q2},e_{q3}$. Further on, we can select one of two ways.

If we are calibrating a low-accuracy IMU, we can neglect the Earth angular velocity, and get the system of 3P+3Q linear equations in the twelve unknowns:
(43)$$\begin{array}{ccc} 0 & = & S_{\omega }{\bar{W}}_{q}+b_{\omega },\;q=1,\ldots ,Q,\\ \omega_{p}e_{1p} & = & S_{\omega }{\bar{W}}_{p}+b_{\omega },\;p=1,\ldots ,P. \end{array}$$

If P≥3,Q≥1, these equations allow us to calculate the elements of $S_{w},b_{\omega }$.

If we are calibrating a high-accuracy IMU, then we must take into account the Earth angular velocity. To get rid of the unknown vectors $e_{2q},e_{3q}$, we multiply (41) with $e_{1q}^{T}$ to get the system of equations
(44)u1=e1qTSωW¯q+e1qTbω,xxxxxxxq=1,…,Q(u1+ωp)e1p=SωW¯p+bω,xxxxxxxu2e2p+u3e3p=SωRp,xxxxxxxp=1,…,P,u2e3p−u3e2p=SωIp.xxxxxxx
This is a system of Q+9P linear equations, with additional unknowns $u_{1},u_{2},u_{3}$, 18 unknowns in total. If P=3,Q=4, which is the minimal number of experiments for calibrating the accelerometers, we get 31 equations—more than enough. Note that we need to know only the absolute value of the Earth angular velocity here—no need to know the azimuth angle or even the latitude.

## 4. Results

The algorithm was tested both with simulated data and with the commercial micro-IMU named x-IMU (Figure 1) from the x-io Technologies Limited company [21]. The goal of simulation was to estimate the potential accuracy of calibration. Since x-IMU is a low-accuracy IMU, the Earth angular velocity was not taken into account in data processing.

### 4.1. Calculating the Fourier Transform with FFT

In the above mathematical formulas, we have assumed infinite duration of experiments; thus, the Fourier transforms of the sine wave and of the constant signal were delta-functions. In the real world, experimental data are recorded at some finite time intervals of duration $T_{p},p=1,\ldots ,P$ for rotations, or $T_{q},q=1,\ldots ,Q$ for static experiments. Thus, the Fourier transform is calculated for a finite support function weighted with some window function h(t)
(45)$$f_{p}\left(t\right)\;\to \;f_{p}^{\prime }\left(t\right)=\left\{\begin{array}{cc} f_{p}\left(t\right)h\left(t\right), & t\in [-T_{p}/2,T_{p}/2],\\ 0. \end{array}\right.$$

It is well known that this provides spectral leakage [20]: the Fourier transform for a windowed exponential $e^{i\omega_{p}t}$ is $H(\omega -\omega_{p})$, where H(ω) is the Fourier transform for h(t). The Fourier transform for windowed $f_{p}^{\prime }\left(t\right)$ is
(46)$$F_{p}^{\prime }\left(\omega \right)=\int_{-\infty }^{\infty }F_{p}\left(\nu \right)H\left(\omega -\nu \right)\;d\nu .$$

The choice of the window h(t) depends on the purpose. For maximal resolution of the spectrum, the rectangular window is the optimum. In this case, H(ω)=sinc(ωT/2) which is a fast oscillating function with minimal width of the central peak. If it is important to avoid false peaks of the spectrum, windows with nearly monotonically decreasing H(iω) are preferred. When the rotation rate is not very stable (as was the case in our experiments, see below), it is better to widen the spectral window so that it would cover the varied main frequencies.

Another consideration arises when we remember that the spectrum is calculated by FFT at discrete set of frequencies with the frequency step $\Delta \omega =2\pi /T_{p}$ [20]. In addition, we need to estimate not just the peak frequency and amplitude, but also the real and imaginary parts of the spectrum which can change fast near the peak frequency. To calculate these values with discrete frequencies, it is preferable to use a window function with a fairly wide central lobe.

As a good compromise, accounting for most of the factors mentioned above, we use here the Hann window function
(47)$$h\left(t\right)=1+\cos\frac{2\pi }{T_{p}}t,\;-\frac{T_{p}}{2}\leq t\leq \frac{T_{p}}{2}.$$

To increase the frequency sampling rate of FFT, we use zero padding [20], adding zeros to the signal on the left and right of the time interval. When *N* zero signal intervals each of duration $T_{p}$ are padded to the signal both on the left and on the right of the recorded data, the frequency sampling rate increases 2N+1 times, to $\Delta \omega =2\pi /T_{p}/\left(2N+1\right)$. To preserve the spectral amplitudes with zero padding, appropriate scaling of the spectrum is done. In the experiments described in the next subsections, we set N=10.

### 4.2. Calibration with Simulated Data

When generating the simulated data, their sampling rate was set to 100 Hz, the shaft rotation main frequency $\omega_{p}$ was chosen as 2.1 rotations per second, the number of rotation experiments was P=3, while the number of static experiments was set to Q=4. In each of the three rotation experiments, one of the axes of the instrument body pointed along the rotation shaft. The generated noise was a Gaussian white noise; RMS of the accelerometer noise was set to $\sigma_{f}=0.01\;{m/s}^{2}$; RMS of the angular rate sensor noise was set to $\sigma_{\omega }=0.1\;\mathrm{rad}/s$. Duration $T_{p}=T_{q}$ of the experiments was varied. The results are shown in Table 1. The errors in $S_{f}$, $S_{\omega }$ are relative, the errors in $b_{f}$, $b_{\omega }$ are in m/s/s and rad/s, respectively. When rotations are done for 10 s at each position (about 20 full revolutions of the body), the calibration accuracy is mediocre. When rotations are done for 80 s (about 160 full revolutions of the body), calibration results become very good. Note that the accuracy of angular rate sensor calibration is considerably higher than that of accelerometer calibration. Probably this is due to using the ellipsoid fitting method to calibrate the accelerometers, which is known not to be the most accurate.

### 4.3. Calibration with Experimental Data

We show here the results of just one test with x-IMU [21], with the data recording rate of 256 Hz. Data recording was done via bluetooth connection. We used a consumer screwdriver as a turntable (Figure 2). This screwdriver is two-speed, the revolution frequency in the slow mode is about 4 rotations per second. The IMU was fastened to the shaft in three different positions: p=1,2,3=P. The static experiments were just the positions of the IMU on the shaft before and after rotations: q=1,2,3,4=Q. No data were collected when the IMU was not fastened to the shaft. All rotations were done clockwise.

We used a factory pre-calibrated IMU. Thus, the raw data we obtained were actually close to the true data; the accelerometer readings were in fractions of *g*, while the angular rate sensor data were in degrees per second. Duration of experiments, including the time taken by mounting and dismounting the IMU on the shaft of the screwdriver, took approximately 15 min, as can be seen from Figure 3.

Figure 3 shows the graph of angular sensor raw reading records during the whole sequence of the experiments. The trigger of the screwdriver was pressed only partially to limit the angular rate of rotations to be within the working range of the IMU angular velocity sensor which is about three rotations per second (see the rope holding the trigger). With the half-pushed trigger, the angular velocity can be seen to be unstable; this fact is seriously decreasing the accuracy of our spectral method, as the main spectral peak at revolution frequency tends to spread wider. Figure 4 shows a fragment of raw accelerometer readings. We see high-frequency oscillations in the data which are probably due to the imperfections of the gearbox. Figure 5 shows a fragment of the FFT of the accelerometer data near the peak at the revolution angular rate of approximately 700 deg/s. We see that the spectrum is spread near the central frequency, partly due to the instability of rotations, partly due to the spectral leakage discussed above [20]. To make the leaked spectrum more smooth, time weighting of the data with the Hann window was done. To increase the sampling frequency in the frequency domain to obtain better estimates of positions of the peaks, the data in time were zero-padded, increasing the length of the data vector 21 times. Figure 6 shows a fragment of FFT of the angular rate sensor data near the zero frequency.

The factory set scaling matrices $S_{f},S_{\omega }$ and these produced by our calibration algorithm for the x-IMU instrument are shown below:$$\begin{array}{ccccc} \mathrm{Factory}:\; & S_{f}= & \left[\begin{array}{ccc} 1 & 0 & 0\\ 0 & 1 & 0\\ 0 & 0 & 1 \end{array}\right],\; & S_{\omega }= & \left[\begin{array}{ccc} 1 & 0 & 0\\ 0 & 1 & 0\\ 0 & 0 & 1 \end{array}\right].\\ \mathrm{Fourier:}:\; & S_{f}= & \left[\begin{array}{ccc} 1.0028 & -0.0008 & -0.0019\\ 0 & 1.0024 & -0.0002\\ 0 & 0 & 1.0033 \end{array}\right],\; & S_{\omega }= & \left[\begin{array}{ccc} 0.9994 & -0.0009 & -0.0017\\ 0.0005 & 0.9987 & -0.0022\\ -0.0079 & -0.0011 & 0.9998 \end{array}\right]. \end{array}$$

The differences in alignment (non-diagonal elements in the matrices) are not more than 0.06 degrees; the differences in scaling factors (diagonal elements of the matrices) are not more than 0.2%. We consider this as a sound verification of our algorithm, since factory calibration of x-IMU has proved to be very accurate in our experiments in pedestrian navigation [10].

We do not discuss here the estimates of the sensor biases since for the micromechanical IMU, the biases are usually fairly unstable and change from switch-on to switch-on.

## 5. Discussion

The paper presented an approach for calibration of a micro-mechanical IMU using a simple turntable, such as a domestic screwdriver, with a calibration algorithm based on the Fourier transform of the data. Verification of our calibration algorithm was done in two ways. First, we used the simulation data and showed that when the duration of experimental rotations was taken more than 40 s (80 revolutions at two rotations per second), the calibration accuracy became sufficient for any purpose. When duration of rotations was below 10 s (20 revolutions at two rotations per second), calibration results were inaccurate. The latter can be explained by the effect of “spectrum leakage” when using FFT on a short time interval.

Second, we performed experiments with the micromechanical inertial measurement unit called x-IMU [21] and compared the results of calibration using our algorithm to the x-IMU factory calibration data, which had proved to be very accurate in our experiments in pedestrian navigation [10]. The differences in the scale factors were around 0.2%, the errors in estimating directions of sensitivity axes were around 0.06 degrees. However, the calibration accuracy was not as high as was expected from simulations. We see the reason for this in that, in our experiments, the rotation angular velocity drifted considerably due to instability of screwdriver RPM. We expect the results to be much better when we use a better turntable than our screwdriver. Another way to increase accuracy is to use reverse rotations, which we did not do.

Some rough estimates not included here show that the accuracy of our algorithm is considerably lower than that of the well-known Kalman filter approach to calibration on a rotating turntable [3], which can be seen to be asymptotically optimal under the assumption of the sensor errors being Gaussian white noise. Nevertheless, a big advantage of our algorithm, which comes from purely algebraic formulas used, is its guaranteed convergence without any need for an initial guess. Without such a guess, the Kalman filter algorithm often diverges. The results of calibration with our algorithm can be used as an initial guess for the Kalman filter algorithm. Joining together the Fourier approach and the Kalman filter approach, with an initial guess provided by the former, is an aim of our future work.

Our algorithm has two versions—for high-grade and low-grade IMU. For the case of low-grade IMU, which is discussed in this paper in more detail, we do not account for the Earth angular velocity, and the algorithm becomes much simpler. Note that when rotation rates become about one turn per second, the Earth angular velocity influence on estimating the gyro scaling matrices is negligible since the rotation rates are several orders of magnitude higher than the Earth rotation rate of 13 degrees per hour. We think that the high-grade IMU version of our algorithm should be used only when a turntable with fairly constant rotation rate is used.

One of the popular micromechanical IMU calibration methods is the so-called calibration without external equipment [13,14], etc., which we have used a lot in our experiments in pedestrian navigation [9]. We have found (the results are not included here) that the new algorithm performs considerably better when it comes to the angular rate calibration of the sensors because their scale factors are estimated better with relatively fast and lengthy rotations, which are impossible to perform by hand. Moreover, contrary to our algorithm, the rotation-by-hand angular rate calibration requires an initial guess on the parameters. Accuracy of accelerometer calibration for the two approaches is comparable. We note also a general advantage of the spectral approach which is that the spectrum provides additional information about the IMU, helping to find out certain effects such as proof mass separation the initial mathematical calibration model has not accounted for.

As already mentioned in the Introduction, we have found just one paper [18] about IMU calibration by applying the Fourier technique which is in relation to ours. The big difference is that in Ref. [18], the true angular velocity and the true specific force are assumed known from the turntable data; thus, the equations are completely linear and can be solved without iterations both in the time domain and in the spectral domain. In our case where the true sensor readings are unknown, the initial equations are nonlinear, which makes the spectral approach we propose look special from the point of view of guaranteed convergence.

## Figures and Tables

**Figure 1 sensors-23-01045-f001:**
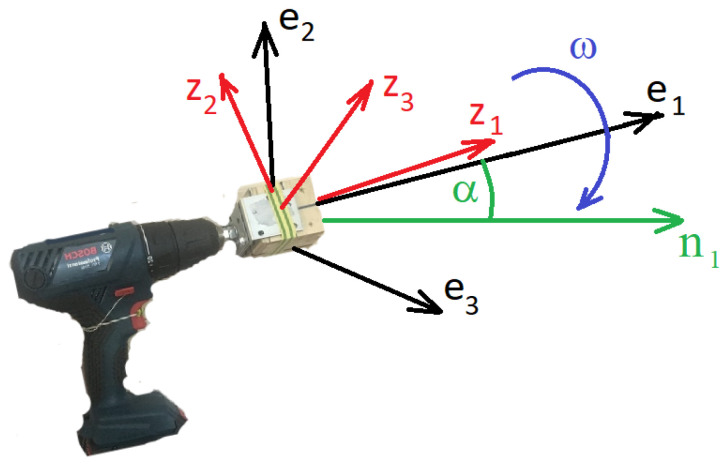
Coordinate frames. The axes $e_{1},e_{2},e_{3}$ form the shaft frame; the axes $z_{1},z_{2},z_{3}$ form the IMU instrument frame. The axis $n_{1}$ is horizontal and lies in the vertical plane of the screwdriver shaft and diverges by the angle ψ from the East direction, the axis $n_{3}$ points up. The angle α is the angle of the shaft to the horizon.

**Figure 2 sensors-23-01045-f002:**
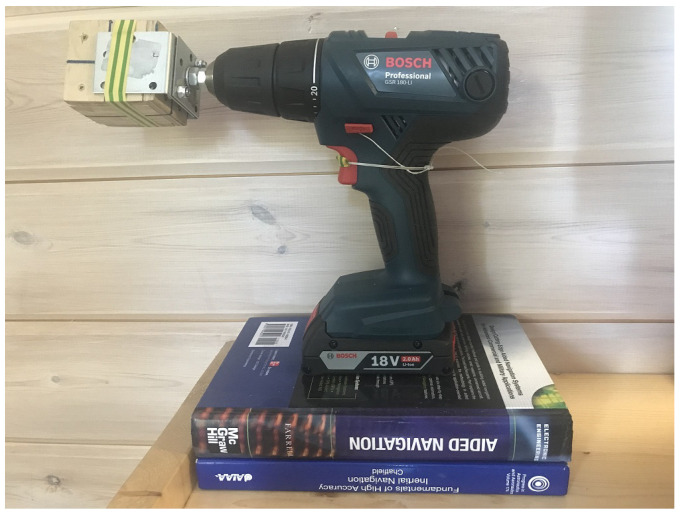
The IMU mounted on the shaft of a screwdriver. The IMU is inside a small wooden box which is fixed to the shaft in several positions by the adhesive tape. The trigger of the screwdriver is held by a rope in the partly pushed position to limit the angular rate of rotations (otherwise, the gyro readings are saturated). Unfortunately, for the partly pushed trigger, the low angular rates of the screwdriver we use are fairly unstable. Books on inertial navigation are of great help in the experiments.

**Figure 3 sensors-23-01045-f003:**
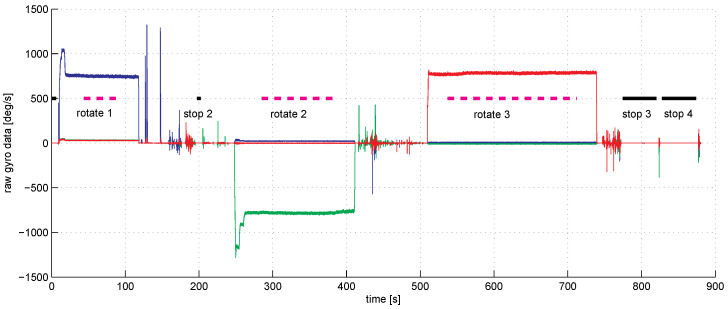
The raw gyro data $w_{p}\left(t\right)$ for p=1 in degrees per second in time (red—axis 1, blue—axis 2, green—axis 3). The IMU was put in three positions on the shaft. In each position, three rotation experiments and four standing still experiments were done. The thick magenta line marks the duration of the rotation experiments. The thick black lines mark the duration of standing still experiments. Note that the angular velocity drifts with time during rotations due to the screwdriver imperfection.

**Figure 4 sensors-23-01045-f004:**
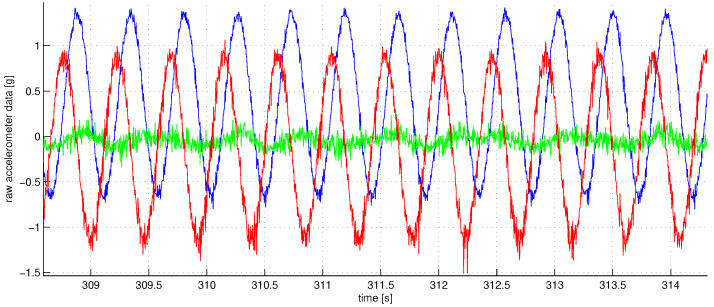
A fraction of the accelerometer data $f_{p}\left(t\right)$ for p=1 in fractions of *g* in time (red—axis 1, blue—axis 2, green—axis 3). High-frequency oscillations we see in the data are caused mainly by the imperfections of the screwdriver bearings and gears. High amplitudes up to 1.5 *g* are caused by the centrifugal forces.

**Figure 5 sensors-23-01045-f005:**
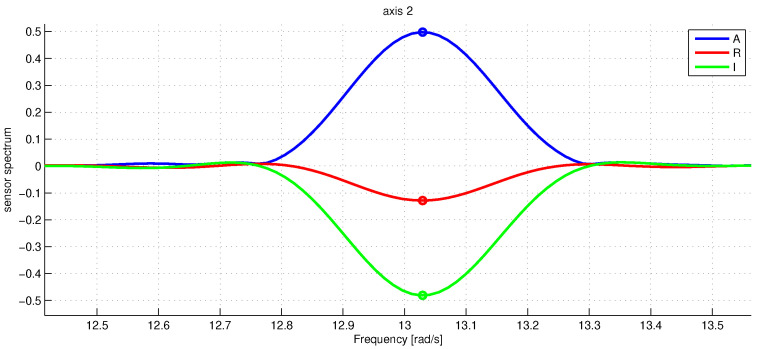
The raw accelerometer data spectrum $F_{p}\left(\omega \right)$ for p=1 at the zero frequency. The blue line is the absolute value of $A_{p}\left(\omega \right)$, the red and green lines denote the real and imaginary parts of $A_{p}\left(\omega \right)$. The marker shows the estimated amplitude peaks. The frequency along the x-axis is in rad/s.

**Figure 6 sensors-23-01045-f006:**
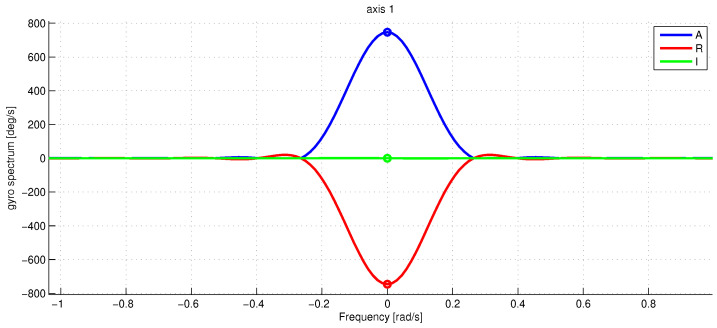
The raw gyro data spectrum $W_{p}\left(\omega \right)$ for p=1 near the zero frequency. The spectrum is normalized in such a way that the peak amplitude ${\bar{W}}_{p}$ is the rotation frequency in deg/s. The blue line is the absolute value of $W_{p}\left(\omega \right)$, the red and green lines denote the real and imaginary part of $W_{p}\left(\omega \right)$. The marker shows the estimated amplitude peaks. The frequency along the x-axis is in rad/s.

**Table 1 sensors-23-01045-t001:** Calibration errors depending on duration $T_{p}=T_{q}$ of each experiment. The errors in $S_{f}$, $S_{\omega }$ are relative, the errors in $b_{f}$, $b_{\omega }$ are absolute, in m/s/s and rad/s, respectively.

$T_{p}$	$S_{f}$ Error	$b_{f}$ Error	$S_{\omega }$ Error	$b_{\omega }$ Error
10 s	0.004	0.05	0.0007	0.006
20 s	0.002	0.03	0.0005	0.004
40 s	0.001	0.02	0.0003	0.002
80 s	0.0007	0.015	0.0002	0.001

## Data Availability

Not applicable.

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
