# Peer review of "Turntable IMU Calibration Algorithm Based on the Fourier Transform Technique"

_sensors, 2023, doi:10.3390/s23021045_

Round 1

Reviewer 1 Report

Comments to the Author:

The manuscript “Turntable IMU calibration algorithm based on the Fourier transform technique” presented an approach to calibration of a mechanical IMU using a simple turntable with calibration algorithm based on the Fourier transform of the data. In general, the structure of this manuscript is reasonable but still need to be improved. The reviewer has some comments for further improving this paper:

1. Please pay attention to the format of the article. The number of consecutive references should be limited. This can be done by mentioning 1 or 2 phrases per reference to show how it is different from the others and why it deserves mentioning.

2. Please explain the direction change in the last paragraph of 3.2.

3. It is recommended to add units to the data in Fig.3 and Fig.4.

4. Could the authors explain how long it took to calibrate with the experimental data, 80 seconds or more?

5. The conclusion part could be more refined to make the findings and contributions of the paper clearer.

Author Response

1. Please pay attention to the format of the article. The number of consecutive references should be limited. This can be done by mentioning 1 or 2 phrases per reference to show how it is different from the others and why it deserves mentioning.

Thank you, will be done.

2. Please explain the direction change in the last paragraph of 3.2.

Thank you, the paragraph is indeed bad. The following text will be added: The algorithm assumes all rotations are clockwise (because positive frequency peak in the Fourier transform is taken). If rotation was actually counter-clockwise, negative frequency peak should be taken.

3. It is recommended to add units to the data in Fig.3 and Fig.4.

 Thank you, the x-axis label will be changed to 'time [s]'. The y-axis data is dimensionless (before calibration), thus no  label seems appropriate.  

4. Could the authors explain how long it took to calibrate with the experimental data, 80 seconds or more?

 Thank you. will add: Calibration experiment took 17500 seconds (appr. 30 minutes).  

5. The conclusion part could be more refined to make the findings and contributions of the paper clearer. 

Thank you, will try to improve it a bit.

Reviewer 2 Report

The paper presented an approach to calibration of a mechanical IMU using a simple turntable like a domestic screwdriver, with calibration algorithm based on the Fourier transform of the data. The new approach is easier to realize and more effective. There are several problems to be solved.

1.     In abstract, the calibration results can be described by some number.

2.     There are too many formulas, suggestions to simplify and enhance the logic to make it easier for readers to understand, and there are too many unknowns, is there a correlation between them?

3.     When the FFT algorithm is adopted, the number of points of the sampled data is generally required to be the power times of 2. When the number of points does not meet the requirements, FFT calculation can be carried out by zero padding, but it will cause the absolute amplitude reading (such as 0Hz component) on the amplitude-frequency characteristic diagram to be inconsistent with the actual signal. In this paper, zero padding method is adopted. What effect will it have on the calibration? How is it considered?

4.     I noticed that many of the formulas were not numbered in the paper. All formulas should be numbered, that will make it easier to find it. Please check all formulas.

5.     The simulated data also give two figure about the accelerate data and the gyro data like Fig.3 and Fig.4. In Table 1, why the Sf error of rotation by 40s is greater than that of rotation by 20s?

6.     Page 13, the Sw of the Fourier result, the number -0.9987 is not close to 1.

7.     What is studied in this paper is the calibration method of bias and scale factor. The simulation data experiment gives the estimation of bias, why the actual x-IMU data experiment does not give the estimation of bias?

8.     In Fig. 3to Fig.6, the unit and title of axis should be marked.

Author Response

1.     In abstract, the calibration results can be described by some number.

The numbers depend on the IMU that is calibrated, and on the required accuracy. It is hard to find a number appropriate for the abstract.

2.     There are too many formulas, suggestions to simplify and enhance the logic to make it easier for readers to understand, and there are too many unknowns, is there a correlation between them?

Thank you, we also feel that there are rather many formulas. But the manuscript is about the algorithm, thus we tried to present explicit formulas that allow the reader to write the code. If we omit some formulas, there will be gaps in the algorithm.

3.     When the FFT algorithm is adopted, the number of points of the sampled data is generally required to be the power times of 2. When the number of points does not meet the requirements, FFT calculation can be carried out by zero padding, but it will cause the absolute amplitude reading (such as 0Hz component) on the amplitude-frequency characteristic diagram to be inconsistent with the actual signal. In this paper, zero padding method is adopted. What effect will it have on the calibration? How is it considered?

We don't require zero padding to power of 2, because the software we use (Matlab and Octave) can calculate FFT for any length of data. We use zero padding to make frequency resolution in the Fourier transform higher. In experiments, we zero pad 20 intervals, and the FFT is rescaled. We shall mention this in the manuscript.

4.     I noticed that many of the formulas were not numbered in the paper. All formulas should be numbered, that will make it easier to find it. Please check all formulas.

We number only the formulas that are referred to later. We think it is the common practice.

5.     The simulated data also give two figure about the accelerate data and the gyro data like Fig.3 and Fig.4. In Table 1, why the Sf error of rotation by 40s is greater than that of rotation by 20s?

1) With simulated data, the gyro readings are just "constant+white noise", the accelerometer readings are just "sine wave+white noise". We don't feel it necessary to present relevant graphs. 
2) Thanks a lot, very bad mistake! We entered the numbers into the Latex file manually and erred, the correct number for 40 seconds is 0.001.

6.     Page 13, the Sw of the Fourier result, the number -0.9987 is not close to 1.

Thanks again, another very bad mistake! The sign must be +.

7.     What is studied in this paper is the calibration method of bias and scale factor. The simulation data experiment gives the estimation of bias, why the actual x-IMU data experiment does not give the estimation of bias?

For IMUs the bias usually changes from switch-off to switch-on, thus it not considered as a result of calibration.   We estimate the bias only because it is coupled with scale factors and misalignment in our algorithm.

8.     In Fig. 3to Fig.6, the unit and title of axis should be marked.

Thank you, we shall mark the x-axis as "time [s]" in Figs. 3-4 and "frequency [rad/s]" in figs 5-6. But the y-xis in these figures is for the raw data, which is just numbers with no dimension.